# Inference technique for the synaptic conductances in rhythmically active networks and application to respiratory central pattern generation circuits

Yaroslav Molkov[1]*, Anke Borgmann[2], Hidehiko Koizumi[2], Noriyuki Hama[2,3], Ruli Zhang[2], Jeffrey Smith[2]

[1]Department of Mathematics and Statistics, Neuroscience Institute, Georgia State University, Atlanta, United States; [2]Cellular and Systems Neurobiology Section, NINDS, Bethesda, United States; [3]Department of Neural and Muscular Physiology, Shimane University School of Medicine, Matsue, Japan

## eLife Assessment

This work describes an inference technique for extracting information about relative contributions of excitatory and inhibitory synaptic drive onto single neurons in neural networks. The electrophysiological techniques and results are of high quality, and the analytical work is novel and potentially powerful, yet with several untested assumptions underlying the approach. This is nevertheless **solid** work that will be **valuable** to neuroscience labs interested in exploring alternative approaches to studies of integrated synaptic connectivity.

**Abstract** Unraveling synaptic interactions between excitatory and inhibitory interneurons within rhythmic neural circuits, such as central pattern generation (CPG) circuits for rhythmic motor behaviors, is critical for deciphering circuit interactions and functional architecture, which is a major problem for understanding how neural circuits operate. Here, we present a general method for extracting and separating patterns of inhibitory and excitatory synaptic conductances at high temporal resolution from single neuronal intracellular recordings in rhythmically active networks. These post-synaptic conductances reflect the combined synaptic inputs from the key interacting neuronal populations and can reveal the functional connectome of the active circuits. To illustrate the applicability of our analytic technique, we employ our method to infer the synaptic conductance profiles in identified rhythmically active interneurons within key microcircuits of the mammalian (mature rat) brainstem respiratory CPG and provide a perspective on how our approach can resolve the functional interactions and circuit organization of these interneuron populations. We demonstrate the versatility of our approach, which can be applied to any other rhythmic circuits where conditions allow for neuronal intracellular recordings.

## Introduction

Rhythmic neural circuits, which are ubiquitous in nervous systems (***Buzsáki, 2006***), generate spatio-temporal patterns of activity by synaptic interactions between neuronal populations, and unraveling the synaptic inputs to circuit neurons at high temporal resolution in the active networks is a major experimental problem in neurobiology for understanding circuit functional dynamics and architecture. Solving this problem ultimately requires electrophysiological methods to extract dynamic patterns of

*For correspondence:
ymolkov@gsu.edu

synaptic conductances in post-synaptic neurons from intracellular recordings during network activity. These patterns often involve temporally complex excitatory and inhibitory synaptic inputs, and thus, methods to extract these inputs must be able to delineate the excitatory and inhibitory components, which can be superimposed during circuit activity, complicating the problem. This study presents a robust and versatile method to extract and separate inhibitory and excitatory synaptic conductance patterns from intracellular recordings. The technique is applicable to any periodically active network where the oscillation period greatly exceeds the interspike intervals and membrane time constant of individual neurons. A synaptic conductance pattern refers to its temporal profile within a single network oscillation cycle. We designed our method to be compatible with various experimental setups, including current-clamp and voltage-clamp protocols, by focusing on the fundamental principles of electrophysiology and network dynamics. Our technique ensures high-resolution temporal profiling of synaptic inputs, making it adaptable to a wide range of rhythmically active neuronal circuits such as vertebrate and invertebrate motor pattern generation (CPG) networks (e.g., *Calabrese and Marder, 2025*), and cortical circuit rhythms (*Yuste et al., 2005*). This generalizability highlights the potential of our approach to provide new insights into the functional organization and dynamics of diverse rhythmic neural circuits. Our approach extends previously proposed methods for extracting dynamic patterns of input synaptic conductances from intracellular recordings from neurons in a variety of circuits (*Borg-Graham et al., 1998*; *Anderson et al., 2000*; *Shu et al., 2003*; *Berg et al., 2007*; *Endo and Kiehn, 2008*; *Wright Jr and Calabrese, 2011*) by providing nearly continuous, high-resolution readouts of excitatory and inhibitory synaptic conductances that are essential for deciphering circuit functional interactions.

To illustrate the utility of our technique, we applied this to circuits in the mammalian brainstem respiratory CPG, which is the fundamental rhythmic neural system that generates and controls breathing movements that are critical for homeostatic regulation of oxygen, carbon dioxide and pH in the brain and body (*Feldman and Smith, 1995*; *Richter and Smith, 2014*). These circuits are continuously active in various experimental preparations, and the experimental accessibility of these circuits provides the opportunity to apply our methods to investigate cellular and network electrophysiological processes underlying rhythmic motor pattern generation in a physiologically important mammalian system (*Richter and Spyer, 2001*; *Smith et al., 2013*). The rhythmic patterns of alternating inspiratory and expiratory phase neuronal activities that coordinate activity of spinal and cranial motoneurons during breathing originate within interacting pontine-medullary excitatory and inhibitory circuits in the brainstem (*Cohen, 1979*; *Bianchi et al., 1995*; *Feldman and Smith, 1995*; *Richter, 1996*; *Smith et al., 2013*). We have targeted specific respiratory microcircuits in the medulla of mature rat brainstemspinal cord preparations in situ containing a variety of neuronal phenotypes and synaptic interactions that are key to the operation of the respiratory CPG. This allowed us to perform single neuron intracellular recordings from various neuronal types in different microcircuits, test different recording protocols to assess the robustness of our approach, and provide examples of how our technique can extract complex patterns of synaptic conductances at high temporal resolution in different types of rhythmic neurons. We illustrate how the synaptic conductance profiles throughout the respiratory cycle imply a 'functional connectome' of respiratory circuit interactions, demonstrating the efficacy of our technique to provide novel information on circuit dynamic organization and operation.

## Results

The general theoretical aspects of our approach for analyzing neuronal excitatory and inhibitory conductances from intracellular recordings in rhythmic circuits, and their specific application to the rhythmically active respiratory circuits, are delineated below. Our objective was to extract synaptic conductance dependence on the phase of the network activity cycle (hereinafter referred to as *synaptic conductance profiles*), representing nearly continuous readouts of excitatory and inhibitory synaptic inputs from the neuronal recordings throughout a cycle of periodic neural activity—the respiratory cycle in our illustration. Any periodic activity can be analyzed in the same way that we describe in this illustration below.

### Reconstruction of synaptic inputs from intracellular recordings

The reconstruction of synaptic conductance profiles is based on the current balance equation:

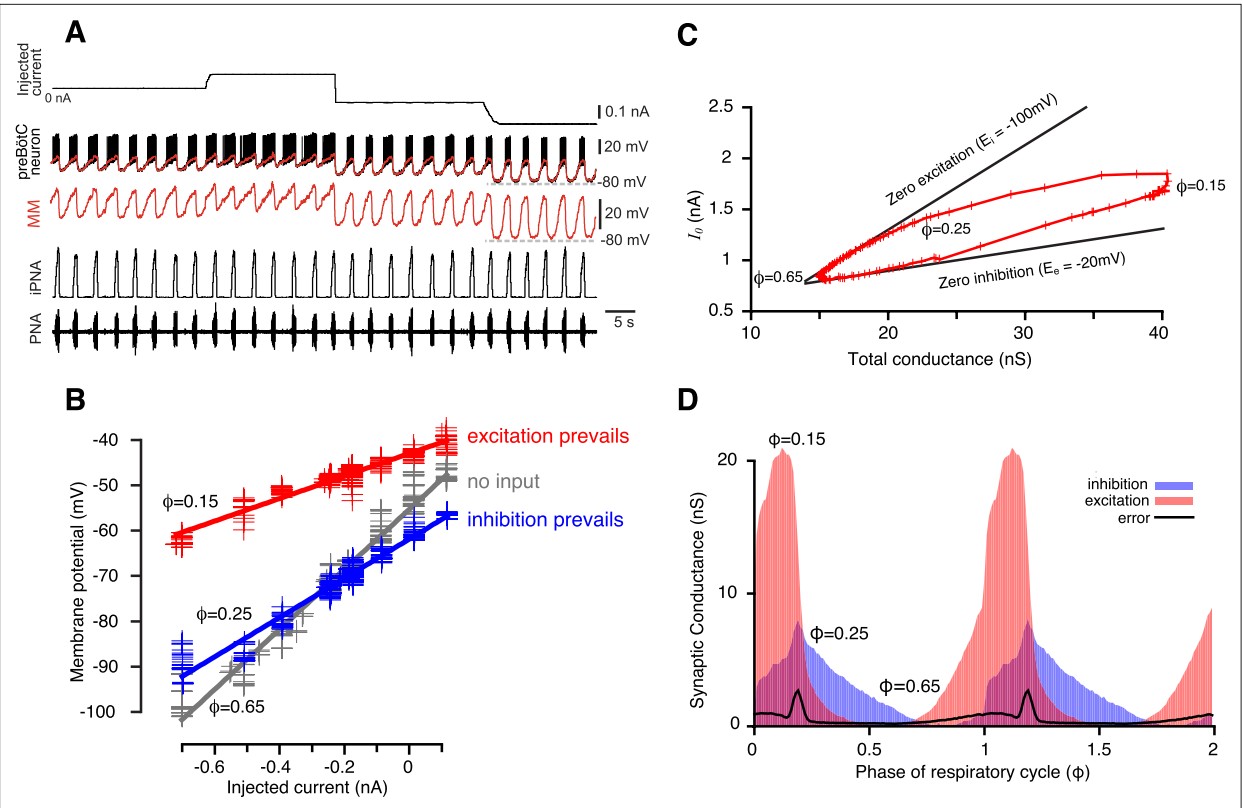

**Figure 1.** Reconstruction of synaptic conductance profiles. (**A**) An example of the intracellular recording from a rhythmic inspiratory neuron (second trace from top) during the stepwise current injection (top trace) protocol, and we primarily used data at hyperpolarized voltages (−100 to −60 mV) for our analyses. Extracellular recording of phrenic nerve activity (PNA, bottom trace) and integrated PNA (iPNA, fourth trace from top) are used for reference. The red trace shows a moving median (MM) filtration with a 0.1 s window of the voltage at higher voltage resolution in the third trace. The MM filters out the spikes while preserving slow voltage waves. (**B**) Plots of injected current vs. membrane potential of a preBötC inspiratory neuron corresponding to three selected values of the respiratory cycle phase (φ). Straight lines of the same colors show the best linear fits. The parameters of the fits–the slope and y-intercept–are used to estimate the total resistance and the resting potential, respectively, for each phase of the cycle. (**C**) A typical example of the wedge diagram. Red plusses represent parameters of linear regressions for 100 phase values. The x-coordinate of each point is the total conductance $G\left(\varphi\right) = 1/R\left(\varphi\right)$, and the y-coordinate is $I_0\left(\varphi\right)$ (see *Equations 2; 11*). Thick black lines are zero excitation (upper boundary) and zero inhibition (lower boundary) lines, respectively (see *Equations 13; 14*). The slopes of the lines correspond to the reversal potentials for inhibition and excitation. (**D**) Calculated dynamic components of synaptic conductances from the same recording using *Equations 6–9* as functions of the phase of the respiratory cycle. Two cycles are shown with integer values of the phase (0, 1, 2) corresponding to a transition from expiration to inspiration. The excitatory synaptic conductance is shown in red, and the inhibitory conductance is shown in blue.

The online version of this article includes the following figure supplement(s) for figure 1:

**Figure supplement 1.** Linear regressions of the injected current vs membrane voltage for different phases of the respiratory cycle.

$$C\frac{dV_m}{dt} = g_{leak}\left(E_{leak} - V_m\left(t\right)\right) + g_i\left(t\right)\left(E_i - V_m\left(t\right)\right) + g_e\left(t\right)\left(E_e - V_m\left(t\right)\right) + I_{inj}\left(t\right) \qquad (1)$$

where $C$ is the membrane capacitance, $V_m$ is the membrane potential, $g_{leak}$ and $E_{leak}$ are the conductance and the reversal potential of the leak current, respectively, $g_i\left(t\right)$ and $g_e\left(t\right)$ are inhibitory and excitatory conductances (subject to retrieval), $E_i$ and $E_e$ are reversal potentials of inhibitory and excitatory synaptic currents, and $I_{inj}\left(t\right)$ is the injected current. We assume that the membrane potential of the cell is in instantaneous equilibrium ($dV_m/dt = 0$), which implies that the membrane time constant $C/g_{leak}$ is much shorter than the time scale of the synaptic conductance variations. This assumption defines the time resolution of our method. For example, $g_{leak}/C$ ratio in the respiratory interneurons was found to be in the range of 80–120 pS/pF (*Koizumi and Smith, 2008*), corresponding to the membrane time constant values of about 10 ms. Synaptic inputs originate from the periodically active network. The respiratory cycle period in our in situ rat brainstem-spinal cord preparation typically

spans approximately 3–4 s (*Figures 1 and 2*), yielding a time resolution of roughly 1/300 of the period. This approach to determining time resolution can be similarly applied to other systems.

Another important assumption is that *Equation (1)* does not include voltage-dependent currents. This means that the equation is simplified to only consider the passive properties of the membrane, such as resistance and capacitance. To ensure that voltage-dependent currents are minimized during the recording, a hyperpolarizing current injection protocol is employed. This involves injecting a negative current into the neuron, which drives the membrane potential to a more negative value. In our test recordings, the membrane potential was typically held between –100 and –60 mV, and we primarily analyzed the data for these hyperpolarized voltages. At this hyperpolarized state, most voltage-gated ion channels are closed and do not significantly contribute to the neuronal current-voltage (I-V) relationships and affect interpretations of our conductance analyses. We note that there may be hyperpolarization-activated currents in the neurons studied, and in current-clamp recordings, non-linearity from such currents, such as h-currents, could introduce voltage-dependent changes in total conductance unrelated to synaptic inputs, potentially skewing the reconstruction. However, our data, as detailed below, consistently demonstrate linear I-V relationships across the voltage range tested for the set of neurons analyzed, indicating a minimal influence from such voltage-dependent currents.

The phase of the respiratory cycle was defined as a piece-wise linear function of time in the following way: $\varphi(t) = (t - t_k) / (t_{k+1} - t_k)$, for times $t$ between the time moments of two consecutive inspiratory phase onsets $t_k$ and $t_{k+1}$. Our approach was based on the idea that in a network exhibiting periodic dynamics, the synaptic conductances depend on the cycle phase, so that $g_{i,e}(t) = g_{i,e}(\varphi(t))$. That is, for a periodic system, specifying that synaptic conductance depends on phase is mathematically equivalent to saying that the conductance depends on time in a periodic manner, and thus, a phase-based representation—where conductances are expressed as functions of the cycle's phase—is a justified and effective approach for capturing their behavior. The respiratory cycle was divided into 100 bins of phase values, and for each bin, the time moments $\{t_{ik}: \varphi(t_{ki}) \in [\varphi_i, \varphi_{i+1}]\}$ that correspond to phases in the bin were collected. For each phase bin $i$ of the respiratory cycle, the relationship between the injected current $I_{inj}(t_{ik})$ and the membrane potential $V_m(t_{ik})$, where $t_k$ are the times that correspond to the phase bin, appears fairly linear (see *Figure 1B*, *Figure 1—figure supplement 1*). Therefore, by linear regression, we can find the slope and $V$-intercept of this dependence in the form:

$$V_m(t_k) = V_0(\varphi) + R(\varphi) \cdot I_{inj}(t_k) + \delta V_m(t_k) \tag{2}$$

where $\varphi = (\varphi_i + \varphi_{i+1})/2$ is the center of the bin. The slope $R(\varphi)$ represents the total resistance, the intercept $V_0(\varphi)$ is the effective resting potential for the phase of the cycle $\varphi$, and $\delta V_m(t_k)$ are random fluctuations of the membrane voltage. Total conductance $G(\varphi)$ was calculated as the reciprocal of the total resistance, $G(\varphi) = 1/R(\varphi)$. The total conductance has static (phase-independent) and dynamic (phase-dependent) components. The latter is assumed to be a sum of dynamic components of the inhibitory and excitatory synaptic conductances. Synaptic conductances can also have static components whose combination cannot be distinguished from a static conductance of any other nature. Accordingly, we can think of the total conductance as a superposition of inhibitory and excitatory conductances whose static components form a static component of the total conductance, and their dynamic components underlie dynamic changes in the total conductance. Hence, the total current can be thought as a combination of the inhibitory and excitatory currents:

$$G(\varphi)(V_m - V_0(\varphi)) = G_i(\varphi)(V_m - E_i) + G_e(\varphi)(V_m - E_e), \tag{3}$$

where $G_i(\varphi)$ and $G_e(\varphi)$ are inhibitory and excitatory synaptic conductances, and $E_i$ and $E_e$ are corresponding reversal potentials. This equation must be fulfilled for any $V_m$, therefore,

$$G(\varphi) = G_i(\varphi) + G_e(\varphi), \tag{4}$$

$$G(\varphi) V_0(\varphi) = G_i(\varphi) E_i + G_e(\varphi) E_e \tag{5}$$

Solving this system of equations for the synaptic conductances yields

$$G_i(\varphi) = G(\varphi)(E_e - V_0(\varphi)) / (E_e - E_i), \tag{6}$$

$$G_e(\varphi) = G(\varphi)(V_0(\varphi) - E_i) / (E_e - E_i) \tag{7}$$

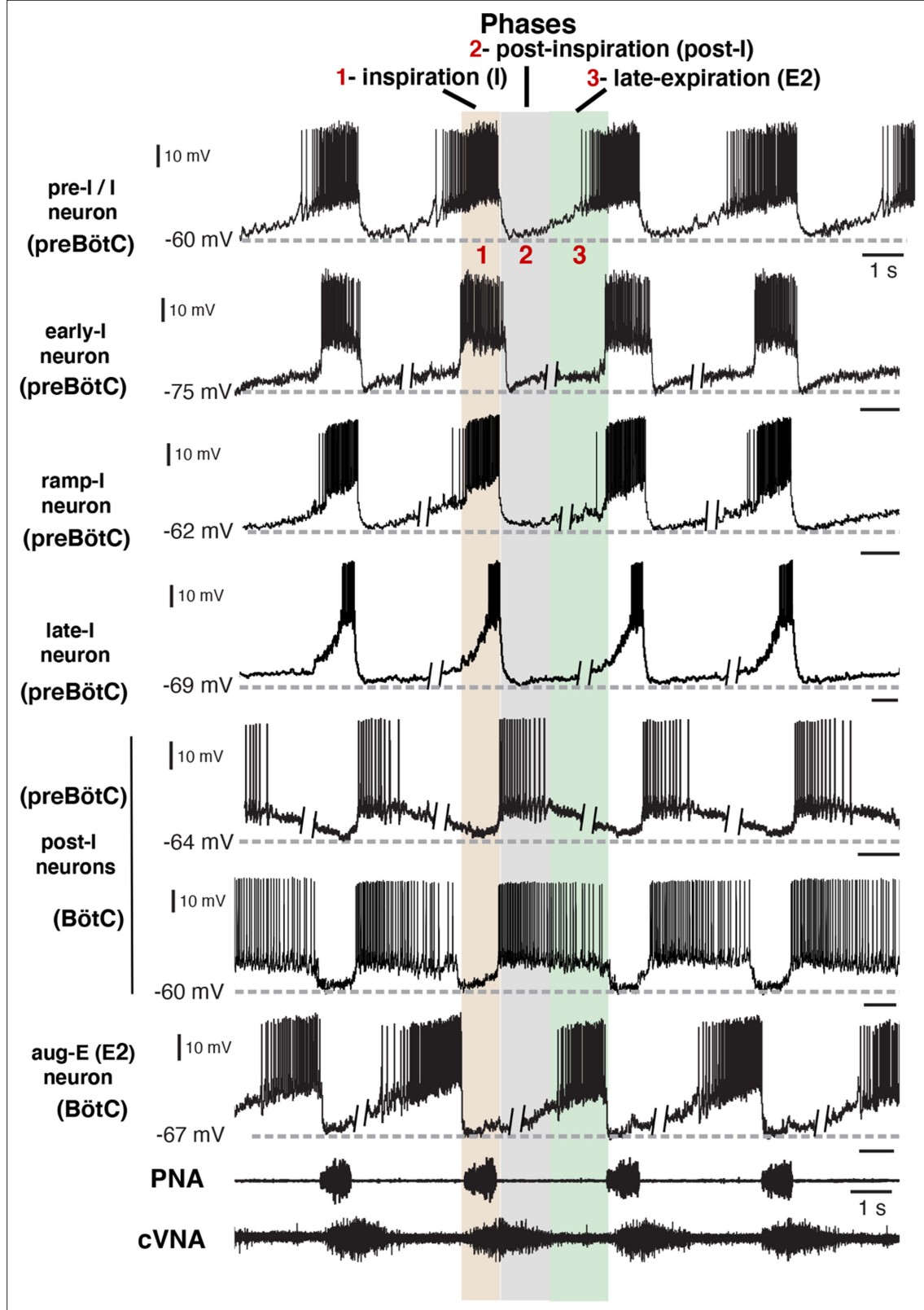

**Figure 2.** Firing patterns of respiratory interneurons. The traces are current clamp intracellular recordings from different types of respiratory neurons distinguished by their action potential firing patterns and activity during different respiratory cycle phases [inspiratory (I), post-I, or late-expiratory/E2 phase]. Extracellular recordings of the phrenic nerve activity (PNA) and central vagus nerve activity (cVNA) are shown below as reference. The nerve recordings were obtained simultaneously with the pre-I/I neuron recording. All other neuron recordings were obtained from different experimental

*Figure 2 continued on next page*

*Figure 2 continued*

preparations and were composited and aligned according to phases of the cycle as indicated at the top. This was done by deleting segments (as indicated) of the membrane potentials during the expiratory phases, since the expiratory intervals largely determine the discharge frequencies which are variable between the in situ preparations, while the inspiratory phase durations are very similar. All neurons included in our study exhibited rhythmic firing patterns consistent with the three-phase organization of the cycle and can be arranged in this format for purposes of illustration. Regions of the ventrolateral medulla (preBötC or BötC) where these example recordings were obtained, as verified histologically (*Figure 2—figure supplement 1*), are indicated.

The online version of this article includes the following figure supplement(s) for figure 2:

**Figure supplement 1.** Respiratory regions of the brainstem ventrolateral medulla and intracellular recording sites.

**Figure supplement 2.** Identification of glycinergic or GABAergic inhibitory neurons.

To find the dynamic components of the synaptic conductances $\Delta G_i\left(\varphi\right)$ and $\Delta G_e\left(\varphi\right)$, we subtract their minimal values over the respiratory cycle:

$$\Delta G_i\left(\varphi\right) = G_i\left(\varphi\right) - \min_\varphi G_i\left(\varphi\right), \tag{8}$$

$$\Delta G_e\left(\varphi\right) = G_e\left(\varphi\right) - \min_\varphi G_e\left(\varphi\right) \tag{9}$$

Finally, we can use the sum of the minimal values found as an estimate for the leak conductance:

$$g_{leak} = \min_\varphi G_i\left(\varphi\right) + \min_\varphi G_e\left(\varphi\right) \tag{10}$$

## Reversal potential estimation

It is possible that during a certain part of the cycle, a neuron receives only one type of synaptic input, inhibitory or excitatory. Below is how this fact can be exploited for obtaining the estimates of reversal potentials for inhibition or excitation. For inhibition, after introducing

$$I_0\left(\varphi\right) = -G\left(\varphi\right) V_0\left(\varphi\right) \tag{11}$$

in (5) and combining (4) and (5), we get:

$$I_0\left(\varphi\right) = -E_i G\left(\varphi\right) - G_e\left(\varphi\right)\left(E_e - E_i\right) \tag{12}$$

Since the total excitatory synaptic conductance $G_e\left(\varphi\right)$ is greater or equal to its static component $G_e^0 = \min_\varphi G_e\left(\varphi\right)$, from (12) it follows that a straight line on the $(G, I_0)$-plane described by

$$I_0\left(\varphi\right) = -E_i G\left(\varphi\right) - G_e^0\left(E_e - E_i\right) \tag{13}$$

serves as an upper boundary of the reconstructed trajectory $\left\{\left(G\left(\varphi\right), I_0\left(\varphi\right)\right), \varphi \in \left[0, 1\right]\right\}$. If we suppose that the neuron receives only phase-dependent inhibition in a certain range of phase, i.e., the excitatory synaptic conductance is equal to its static component in this range, $G_e\left(\varphi\right) = G_e^0$, then the trajectory moves strictly along the line described by (13), which can be referred to as a *zero excitation line*. By identifying the upper portion of the trajectory with nearly linear behavior, we can perform linear regression of this portion to find its slope $-E_i$, and thus estimate the reversal potential for inhibition (see *Figure 1C*). Similarly, for the phases of the cycle where only excitatory input is present, we can derive:

$$I_0\left(\varphi\right) = -E_e G\left(\varphi\right) + G_i^0\left(E_e - E_i\right) \tag{14}$$

where $G_i^0 = \min_\varphi G_i\left(\varphi\right)$ is a static component of the inhibitory conductance and call it the *zero inhibition line*. The latter obviously serves as a lower boundary of the trajectory. Together, the zero excitation and zero inhibition lines form a wedge that bounds the trajectory, so hereinafter we refer to such plots as *wedge diagrams* (*Figure 1C*).

Interestingly, the linear upper boundary of the trajectory was seen in almost all recordings processed, implying that, first, during certain phase(s) of the respiratory cycle, the interneurons indeed receive

only inhibitory input, and second, we could use the identified parameters of the zero-excitation line to measure the inhibitory reversal potential. The same approach might be applied for the determination of the reversal potential for excitation if there was a certain range of phases when a cell received the excitatory input only. Since $E_e$ is expected to be close to 0 mV, this would be clearly identifiable by the presence of a nearly horizontal part of the curve serving as a lower boundary of the same. This kind of behavior was not found in most of the recordings, so, as commonly accepted, $E_e$ was set to –10 mV in those cases. However, as we demonstrate below, variation of reversal potentials over a wide range of possible values does not affect the shape of synaptic conductance profiles qualitatively.

## Firing patterns, synaptic input conductances, and inferred functional connectome of respiratory interneurons: an illustration

For our illustration of extracting synaptic conductances with the above approach, we targeted neurons in key interacting respiratory circuits located in discrete bilateral regions of the ventrolateral medulla, specifically in the preBötzinger complex (preBötC) region, which contains local excitatory and inhibitory circuit neurons critical for generating rhythmic inspiratory activity, and the adjacent more rostral Bötzinger complex (BötC) region (refer to *Figure 2—figure supplement 1*), containing neurons generating rhythmic expiratory activity that interact with preBötC circuits, including by inhibitory circuit connections (e.g. *Lindsey et al., 2012*; *Smith et al., 2013*; *Molkov et al., 2017*; *Ausborn et al., 2018*). Thus, targeting these regional microcircuits allowed us to test our methods for extracting various patterns of excitatory and inhibitory synaptic inputs in two functionally distinct microcircuits. Also, there are various electrophysiological phenotypes of respiratory neurons with different patterns of neuronal spiking activity during the respiratory cycle in these regions, thus providing a diversity of rhythmic neuronal types to correlate with synaptic input patterns from our analysis, which is useful for illustrating the utility of our techniques for delineating various functional synaptic interactions. We have used sharp microelectrode intracellular recordings in mature rat brainstem-spinal cord preparations in situ (see Materials and methods) for our analysis since this approach allowed us to readily record deep in the brainstem in these targeted regions from multiple neurons under current- and voltage-clamp with a single electrode. This provides complementary information to test the robustness of our approach, which should be insensitive to the mode of intracellular recording. Although our techniques can be applied to any type of rhythmic neuron, we specifically targeted and analyzed interneurons, verified by immunochemistry (as described below and in Material and methods), which are the most technically challenging to record from due to small neuronal sizes and potential recording instability, but are functionally critical for analyzing synaptic mechanisms of neural activity pattern generation. Thus, we could test how our technique performs for important neuronal populations under recording conditions that would likely be encountered for many rhythmic interneuronal circuits in nervous systems.

### Firing patterns of respiratory interneurons

Under normal physiological conditions, the neuronal activity during the respiratory cycle consists of three main activity phases [inspiratory (I), post-inspiratory (post-I), and stage 2 or late expiratory (E2)] that are well documented in the literature (e.g. *Lindsey et al., 2012*; *Richter and Smith, 2014*). Previously described respiratory populations based on their specific firing patterns in relation to these different phases of the cycle (see *Figure 2*) include pre-inspiratory/inspiratory (pre-I/I), ramping inspiratory (ramp-I), early-inspiratory (early-I), late inspiratory (late-I), post-inspiratory (post-I), and augmenting expiratory (aug-E, active during E2) neurons. We have recorded neurons from all these populations (*Figure 2*) and have analyzed and presented representative synaptic input patterns (*Figure 3*) for each electrophysiological phenotype. In some cases, as an informative addition to our electrophysiological analyses, we labeled the recorded neurons by intracellular injection of biocytin contained in the electrode recording solution (see Materials and methods) for subsequent histological verification (*Figure 2—figure supplement 1*) of the recording location, interneuron identification, and in some cases neurotransmitter phenotype by immunohistochemistry (n=34 neurons total labeled). The latter enabled us to identify inhibitory neurons (glycinergic or GABAergic) (e.g. see *Figure 2—figure supplement 2*) for five of the neuron types (pre-I/I, ramp-I, early-I, post-I, aug-E) as described below, which aids in circuit analysis and the corroboration of inhibitory inputs from neurons with particular firing patterns.

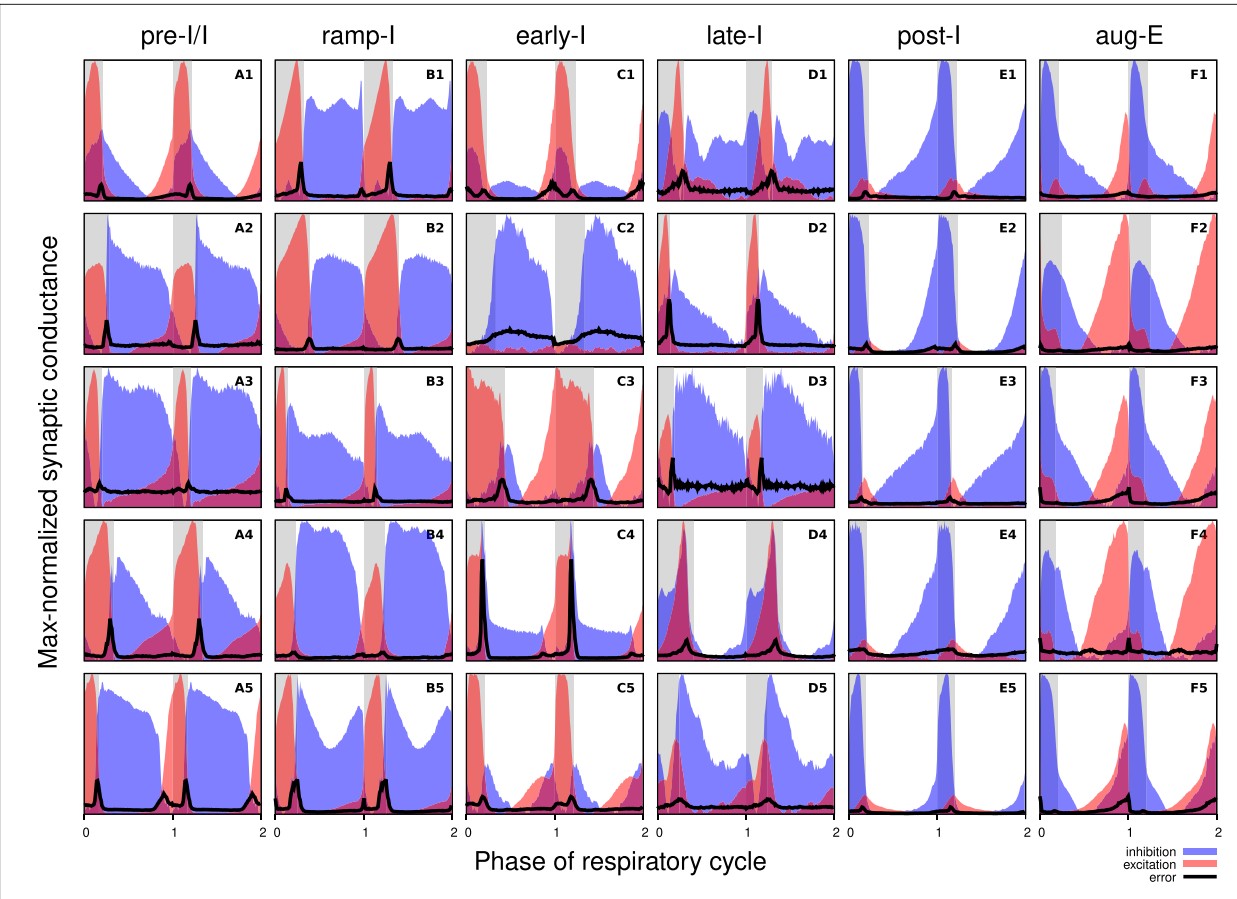

**Figure 3.** Synaptic conductance profiles of major respiratory neuronal phenotypes. Each panel depicts how the dynamic components of excitatory (red) and inhibitory (blue) synaptic conductances, normalized to the maximal value, vary with the cycle phase for an individual neuron. Neuronal firing phenotypes are listed at the top. Two respiratory cycles are shown so that transitions between the respiratory phases are clearly seen. The inspiratory phases are highlighted by gray bars. Integer phase values (0, 1, 2) correspond to transitions from expiration to inspiration as determined by the onset of the phrenic motor output recorded simultaneously. The error for each phase value is indicated by a thick black line in each panel. Each panel is labelled according to the firing pattern of the recorded neuron (A: pre-I/I, B: ramp-I, C: early-I, D: late-I, E: post-I, F: aug-E).

## Synaptic conductances and functional connectome of respiratory interneurons

*Figure 3* shows the synaptic conductance profiles of the major interneuronal electrophysiological phenotypes whose firing patterns are illustrated in *Figure 2*. Within one population of the same electrophysiological phenotype, patterns of synaptic inputs were qualitatively similar, although there was neuron-to-neuron variability in calculated conductance values, and details of the conductance profiles could vary (*Figure 3*) among multiple examples of each type of recorded neurons. From these temporal patterns of synaptic conductances, as explained below, we infer functional connectomes or circuit motifs for each electrophysiological phenotype, as illustrated in *Figure 4*, which is an important outcome of our analyses. In general, these circuit connection motifs are constructed by considering that the different components of the synaptic conductance profiles are formed by convergent inputs from neuronal populations of specific firing phenotypes and, therefore, can be interpreted as reflecting activity patterns of these pre-synaptic populations. By matching these activity patterns with patterns of synaptic conductances in the recorded post-synaptic neurons, we can suggest functional connections between neuronal populations. The information on inhibitory transmitter phenotype for neurons with different firing patterns was also collected and indeed corroborates that subsets of the presynaptic neurons are inhibitory as inferred from the conductance profiles, which gives additional confidence in the correlation between pre-synaptic firing patterns and likely post-synaptic interactions.

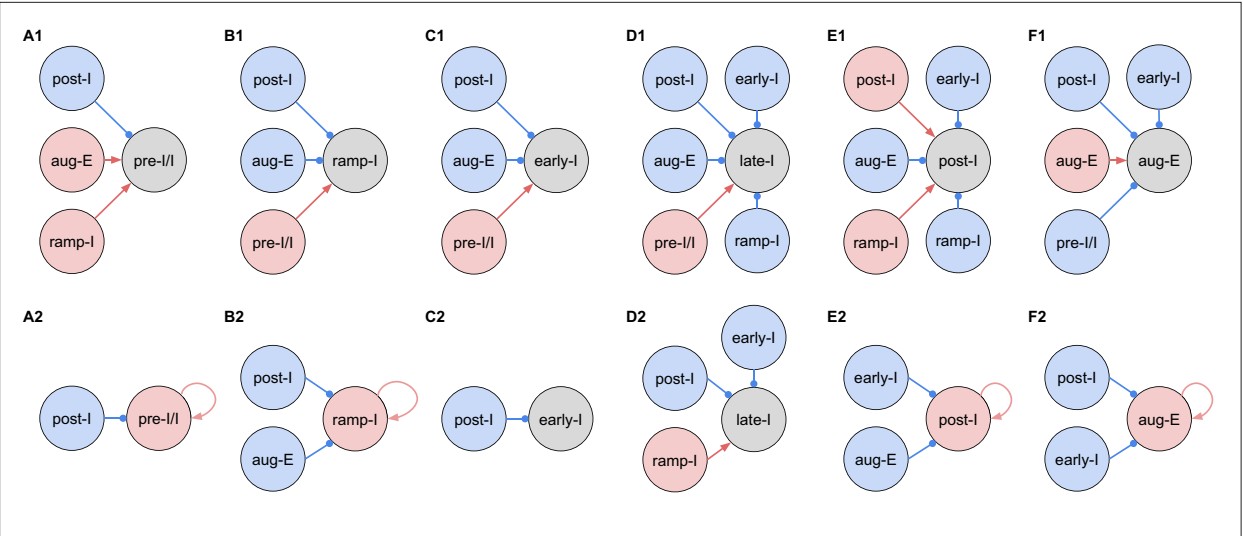

**Figure 4.** Inferred connections between different respiratory interneuronal populations. By matching the firing phenotype of the receiver neuron and its synaptic inputs, we infer possible motifs of interactions between the functional populations of the active network. In this representation, we assume that the functional connection between populations is present, if a post-synaptic neuron of a particular firing phenotype has a dynamic component of the synaptic conductance statistically significantly different from zero (see Materials and methods: Statistical significance) in the phase range corresponding to the activity pattern of the pre-synaptic population. Here, we show two examples of such inferences for each phenotype representing the most straightforward interpretation of synaptic inputs (**A1, B1, C1, D1, E1, and F1**), as well as interactions involving the least number of populations (**A2, B2, C2, D2, E2, and F2**), reflecting the cell-to-cell variability in conductance profiles with some neurons within a given electrophysiological phenotype exhibiting the smaller set of synaptic inputs. Possible inhibitory/excitatory connections are shown by blue/red lines ending with blue circles/red arrows originating from presynaptic neurons and terminating at the post-synaptic neurons. Inhibitory sources are shown in blue, and the excitatory sources are shown in red. These circuit motifs are consistent with the patterns of synaptic inputs to the various neurons, as described in the text for each electrophysiological phenotype, and account for our immuno-histochemical identification of inhibitory neurons as subpopulations of most of the electrophysiological types, as indicated in the text.

The online version of this article includes the following figure supplement(s) for figure 4:

**Figure supplement 1.** Combinatorial circuit schematic.

To illustrate in more detail how we have reconstructed the circuit connection motifs in *Figure 4*, we describe how we interpret and translate the synaptic conductance profiles for each electrophysiological phenotype.

Pre-I/I cells (n=25 neurons recorded in the preBötC) in all cases received decrementing expiratory inhibition and augmenting excitation during late expiration, as well as strong ramping inspiratory excitation (*Figures 3A1–5*). As shown in *Figure 4A1 and a* straight-forward interpretation of this pattern of conductances is that the pre-I/I population receives inhibitory inputs from post-I spiking neurons and excitatory inputs from aug-E and ramp-I populations. Alternatively, since a combination of aug-E and ramp-I excitatory inputs is qualitatively similar to an input from the pre-I/I population itself, it is also possible that excitatory pre-I/I neurons are recurrently interconnected within this population (*Figure 4A2*). These possibilities are not mutually exclusive, and both connectivity patterns could be expressed in the respiratory circuits. We labeled 13 of these pre-I/I neurons during recording, two of which were identified as inhibitory neurons by neurotransmitter immuno-labeling for glycine or GABA (1 glycinergic and 1 GABAergic neuron identified), which is incorporated in *Figure 4F1*.

Neurons with the ramp-I spiking patterns (n=24 recorded in preBötC) also received ramping inspiratory phase excitation, but unlike some pre-I/I cells, they generally showed strong synaptic inhibition throughout the post-inspiratory and late-expiratory (E2) phases (*Figures 3B1–5*). Some of these neurons also received excitatory input during the E2 expiratory/pre-inspiratory phase. Therefore, it is reasonable to suggest that ramp-I neurons receive inhibitory inputs from post-I and aug-E populations, as well as an excitatory input from pre-I/I neurons (*Figure 4B1*). Another possibility is that the excitatory ramp-I neurons not receiving excitation during the pre-inspiratory phase have recurrent excitation within their population (*Figure 4B2*). Four out of seven of these neurons were labeled in the preBötC and identified as inhibitory (glycinergic) neurons (incorporated in *Figure 4D1 and E1*).

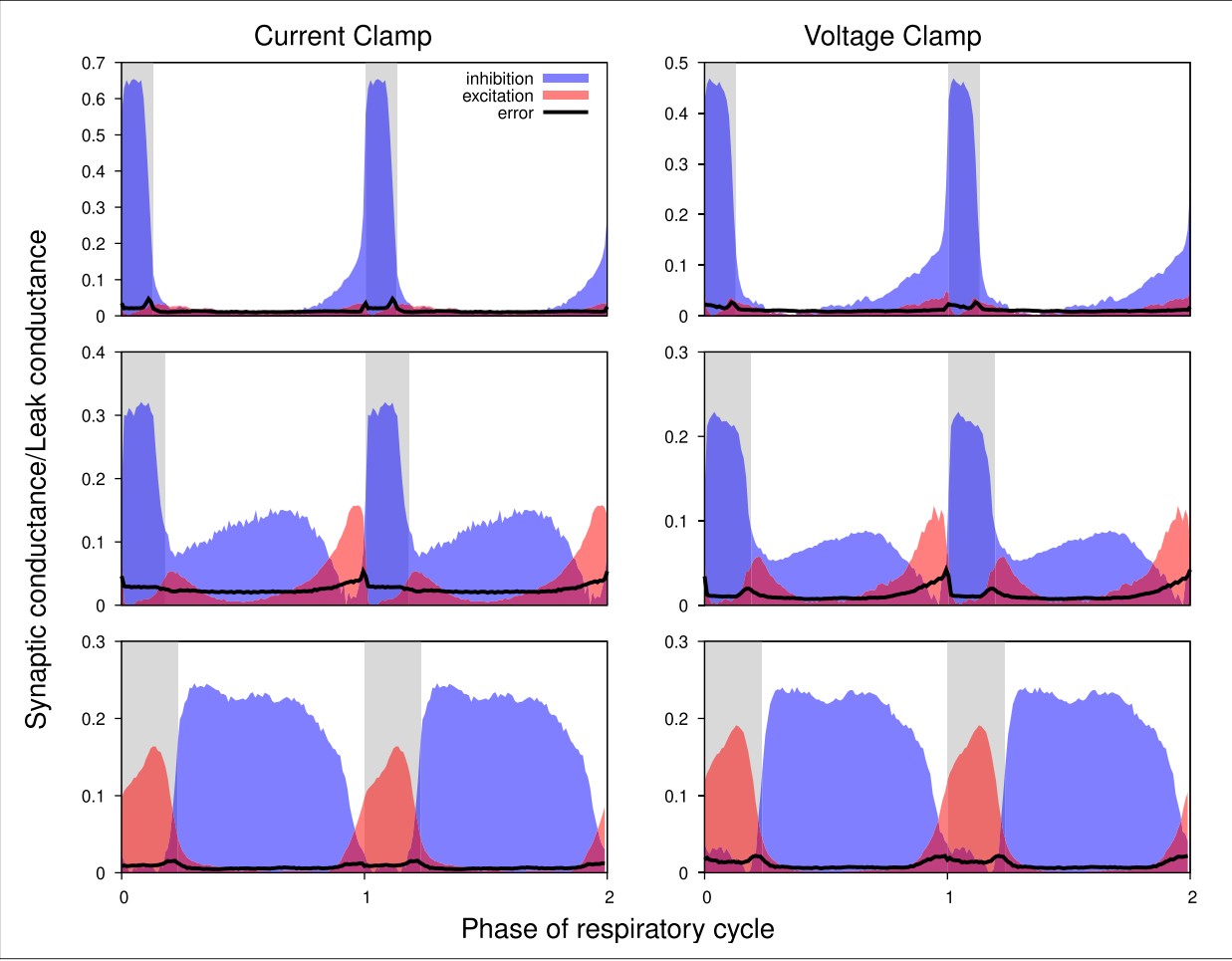

**Figure 5.** Comparison of the synaptic conductance profiles reconstructed from current- and voltage-clamp recordings. Conductance profiles for three representative preBötC neurons are illustrated (from current-clamp recordings at left and voltage-clamp recordings from the same neurons at right). Two respiratory cycles are shown for each neuron (post-I, top; aug-E, middle; ramp-I, bottom panels) with inspiratory phases highlighted by gray bars. Calculated synaptic conductances are normalized by the estimates of the leak conductance. While the absolute conductance values may vary, the shapes of the synaptic conductance profiles, as well as the relationship between the excitatory and inhibitory conductances, are consistent across these recording protocols.

All early-I spiking neurons (n=6 in preBötC) showed decrementing expiratory phase inhibition (*Figures 3C1–5*). Most of these cells also received pre-inspiratory/late expiratory and inspiratory excitation (*Figures 3C1 and –5*), while some of them did not (*Figure 3C2*), which we interpret as shown in *Figure 4C1 and C2*, respectively. The excitatory pre-I/I input can also be a combination of aug-E and ramp-I excitation as in *Figure 4A1*. One of the three labeled early-I neurons was identified as an inhibitory (GABAergic) neuron (*Figure 2—figure supplement 2*), corroborating the early-I inhibitory inputs inferred from conductance profiles shown in *Figures 4 D1 and 2*, *Figure 4E1 and 2*, *Figure 4F1 and 2*.

Synaptic inputs to late-I cells (n=6 in preBötC) were similar to ramp-I neurons (*Figures 3D1–5*) plus concurrent, inspiratory phase inhibition. So, in addition to inhibitory inputs from post-I and aug-E neurons and pre-I/I (in some cases) and ramp-I excitation, they likely receive inhibitory inputs from early-I spiking neurons (*Figure 4D1*). Some of these cells showed neither inhibition nor excitation at the end of expiration (*Figure 3D2*), so they might receive no aug-E inhibition and ramp-I rather than pre-I/I excitation (*Figure 4D2*). These neurons were infrequently recorded, and none were successfully labeled for neurotransmitter type identification.

Post-I spiking neurons (n=40: 25 in preBötC, 15 in BötC) that always exhibit a decrementing firing pattern during expiration received strong decrementing inhibition during inspiration and incrementing inhibition during late expiration (*Figures 3E1–5*). They also showed relatively weak

excitation during post-inspiration, usually accompanied by comparable in magnitude ramping excitation during inspiration. The corresponding circuit elements representing inputs to the post-I neurons are shown in *Figure 4E1*. Since post-I neurons receive post-I excitation, it is possible that excitatory post-I cells form a population with recurrent excitatory interconnections. Besides, some of these neurons did not have any significant excitatory input during inspiration (*Figure 3E3*). The latter possibilities are shown in *Figure 4E2*. Eight of the 20 post-I neurons labeled (five in preBötC and three BötC neurons) were identified as glycinergic (*Figure 2—figure supplement 2*), consistent with a previous study (*Ezure et al., 2003*), and corroborating the post-I inhibitory inputs shown in *Figures 4A1 and 2*, *Figure 4B1 and 2*, *Figure 4C1 and 2* , *Figure 4D1 and 2* and *Figure 4F1 and 2*.

Finally, all the aug-E spiking neurons (n=19 recorded: nine neurons in preBötC, 10 in BötC) received decrementing inhibitory inputs during expiratory and inspiratory phases, as well as an excitatory input that started in the middle of expiration and incremented towards the end of expiration (*Figures 3F1–5*). Some of these neurons received pre-inspiratory inhibition. We interpret this as the aug-E population receiving inputs from early-I, post-I, and pre-I/I inhibitory populations and the excitatory input from aug-E neurons (*Figure 4F1*). Similar to other interpretations, it is also possible that the excitatory aug-E neurons form a population with recurrent excitation with an augmenting pattern that matches their activity profile, while some of these neurons do not receive pre-I inhibition (*Figure 4F2*). Three out of six of these neurons labeled were identified as glycinergic neurons (one preBötC neuron and two BötC neurons) (*Figure 2—figure supplement 2*), as inferred from the synaptic conductance profiles in *Figure 4B1 and 2, C1, D1, E1*.

## Validating reliability of inferred synaptic conductance profiles

### Synaptic conductance profiles from current- and voltage-clamp recordings

The traditional way to measure transmembrane currents is by voltage-clamp intracellular recordings. Using this experimental protocol, the membrane potential of the neuron is held at a constant level, thus maintaining all voltage-dependent membrane currents at their steady states. Therefore, in voltage-clamp conditions, any total current variations result exclusively from time-varying synaptic inputs, while in the current-clamp protocol variations of the voltage can potentially be contaminated by transient dynamics of the voltage-gated channels. However, the implementation of the voltage-clamp protocol is technically far more challenging with sharp high-impedance microelectrodes than injecting a constant current. By design, our technique is indifferent to what strategy is used for the injected current variations, so we performed both current- and voltage-clamp recordings from select neurons (n=10) where the voltage-clamp could be satisfactorily established following the current-clamp recording in each of these neurons and compared the results of the synaptic input interferences as a test of the reliability of our methods.

*Figure 5* shows the excitatory and inhibitory synaptic conductance profiles extracted from current- (left) and voltage-clamp (right) segments of the intracellular recordings for three different types of representative neurons. It is evident that the shapes of the conductance profiles, as well as the relationship between the excitatory and inhibitory conductances, extracted are very similar. Quantitative variations in the amplitude of the synaptic inputs are similar to those observed when using different epochs of the same current-clamp recordings (see below). Thus, we conclude that our synaptic conductance inferences obtained by current clamp recordings favorably match the conductance profiles obtained from direct recordings of synaptic currents during voltage clamp.

### Uncertainties concerned with estimates of synaptic reversal potentials

As noted, it was not always possible to identify linear portions of GI-contours suitable for reliable estimation of the reversal potentials. In addition, the procedure described does not imply an unequivocal way for finding the errors for the reversal potential estimates. Accordingly, for the synaptic input classification purposes, we varied the reversal potentials in the predefined ranges when using *Equations 6; 7* to calculate the synaptic conductances. We used the following ranges: [−20, 0] mV and [−110, −90] mV for the reversal potentials for excitation and inhibition, respectively. *Figure 6* illustrates that although the absolute values of the conductances vary to some extent as we vary the reversal potentials, the shape of their reconstructed profiles remains consistent.

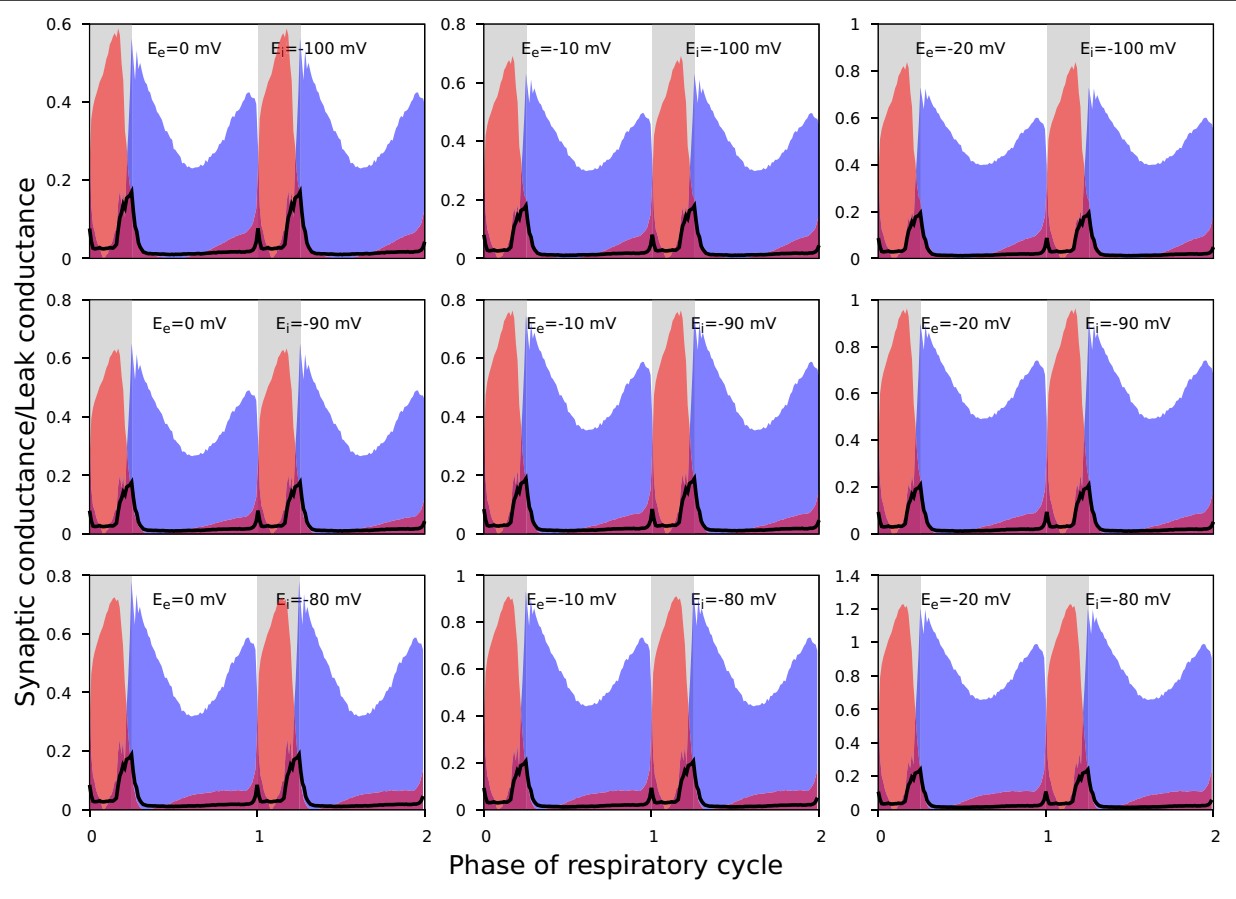

**Figure 6.** The effect of reversal potential variation on the synaptic input reconstruction results. We used the recording of the ramp-I neuron shown in *Figure 3B5* and recalculated its synaptic conductances for different combinations of the inhibitory and excitatory reversal potential values as indicated in each plot. Synaptic conductances are normalized by the estimates of the leak conductance. While the absolute conductance values vary with the reversal potentials, the shapes of the synaptic profiles, as well as the relationship between the excitatory and inhibitory conductances, are consistent across different values of the excitatory and inhibitory reversal potentials used.

### Evaluating the effects of non-stationarity of the intracellular recordings

Our approach implicitly assumes that the intrinsic properties of the recorded neurons do not change (much) throughout the epoch used for the calculations. To verify the plausibility of this hypothesis, whenever possible, we used two different epochs from the same recordings satisfying the criteria formulated in the *Preprocessing* section of the Methods for the complete analysis. We found that differences between the results for different epochs in most cases were comparable to those observed in the recordings obtained by current- and voltage-clamp protocols from the same neuron. *Figure 7* shows the results for two adjacent non-overlapping epochs for three different neurons.

In summary, our evaluation of the major assumptions and potential sources of error with our procedures for extracting temporal patterns of synaptic conductances, including recording procedures and cycle phase-dependent linearity of current-voltage relations (*Figure 1—figure supplement 1*), indicates that the temporal profiles of conductances obtained by our method are reliable for functional circuit reconstruction, as we elaborate further below.

## Discussion

In our analyses of synaptic conductance profiles in rhythmically active neurons from neuronal intracellular recordings, we propose that these profiles represent the convergent synaptic inputs from many neurons, and therefore represent the synaptic interactions of key functional populations operating in the active networks, which provides a basis for functional network reconstruction.

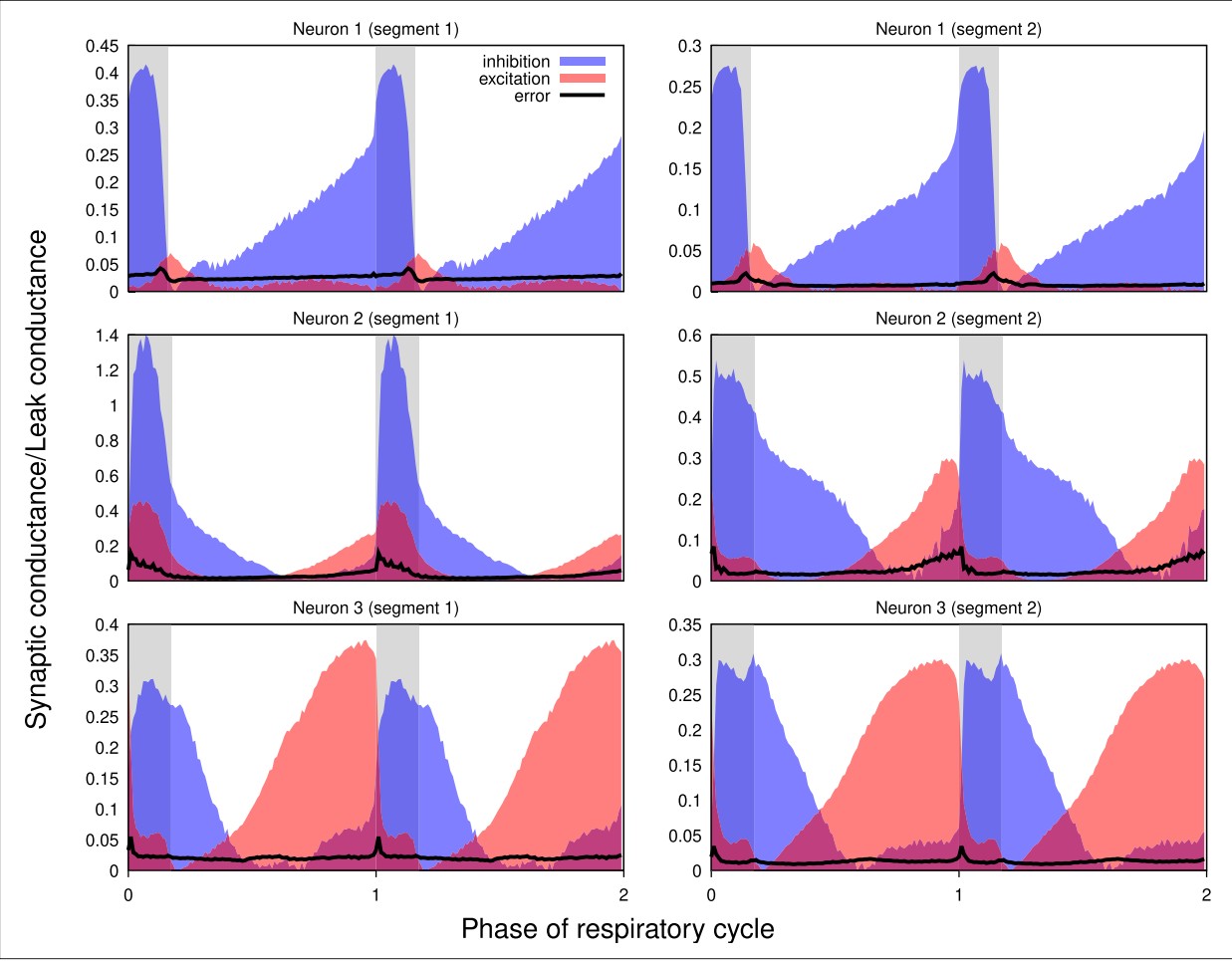

**Figure 7.** Non-stationarity of the recordings does not qualitatively affect the reconstructed profiles of synaptic conductances. Synaptic conductance profiles calculated using non-overlapping segments of the recordings (left and right) from three different neurons (post-I, top; aug-E, middle, and bottom). The similarities of the conductance profiles for different segments reflect the stability of the recordings and neuronal intrinsic properties and patterns of synaptic inputs.

Accordingly, a main objective of this study was to establish the utility and reliability of our technique, based on theoretical analyses and experimental validation, for extracting and separating profiles of inhibitory and excitatory conductances at high temporal resolution from intracellular recording data. We demonstrate the efficiency and robustness of our technique by applying our analyses to infer synaptic inputs to rhythmically active respiratory neurons comprising key micro-circuits within the medullary preBötC and BötC regions of the brainstem respiratory CPG, which provided a diversity of neuronal electrophysiological phenotypes in various rhythmic microcircuits to test the performance of our technique. Our approach extends previously proposed methods for extracting dynamic patterns of synaptic conductances in post-synaptic neurons in various other neural circuits (*Borg-Graham et al., 1998*; *Anderson et al., 2000*; *Shu et al., 2003*; *Berg et al., 2007*; *Endo and Kiehn, 2008*; *Wright Jr and Calabrese, 2011*), by providing higher temporal resolution with nearly continuous readouts of inhibitory and excitatory synaptic conductances, in our illustration throughout the respiratory cycle for different respiratory interneuron types. We have obtained previously unavailable synaptic conductance profiles for the different neuron populations, and illustrate how this analysis can be used to infer functional interactions and circuit configurations of interneuron populations, which demonstrates the utility of our technique. Below, we further address technical issues affecting the robustness of our approach, and then discuss the general applicability to rhythmically active circuits.

## Estimation of reversal potentials and errors in conductance analysis

We have noted above and evaluated uncertainties concerned with synaptic reversal potentials and assumptions about the stability of the cellular properties and recordings on the inferred synaptic conductance profiles, but it is important to elaborate on certain points. We have used an experimental preparation and setup that provides optimal stability for sharp electrode intracellular recordings. We used strict criteria for assessing and accepting the quality and stability of the recordings. For the most part, our analysis results for a given neuron that meets our criteria were reproducible for different recording epochs and conditions. We also generated continuous error estimates for each example shown and confirmed the statistical significance of conductance values. Regarding stationarity, our results suggest that any differences between the inferred conductance profiles for different recording epochs and conditions in most cases were comparable to those between recordings obtained by current- and voltage-clamp protocols.

Most neurons that we have analyzed receive inhibition only during certain parts of the respiratory cycle, regardless of whether they are inspiratory or expiratory neuron types. This allowed retrieving their reversal potentials for inhibition using the approach described in Materials and methods. Interestingly, the inhibitory synaptic reversal potentials retrieved turned out to vary significantly among cells between –62 mv and –116 mV. The reversal potential estimates were in good correspondence with actual reversals of inhibitory events when a cell was hyperpolarized below the reversal potential. Therefore, regarding the analysis, the results were reliable. Nevertheless, on average, reversal potentials were much more negative than what would have been expected. It is well known that in intracellular recordings, the length constant of the neuron and the distance of the microelectrode from the synaptic inputs determine the measured reversal potential (*Calvin, 1969*; *Spruston et al., 1993*; *Wehr and Zador, 2003*). For the precise extraction of inhibitory and excitatory conductances from the total conductance, reversal potential values may be critical. The variability of the inhibitory reversal potential emphasizes the need to determine the reversal potential for each neuron separately. Otherwise, a wrong inhibitory reversible potential value used for the reconstruction may lead to incorrect conductance profiles. For example, a large positive bias in the inhibitory reversal potential will have the effect that a part of the inhibitory conductance may be mislabeled as excitatory conductance and, therefore, show a pattern of concurrent inhibition and excitation. Our method for the estimation of inhibitory reversible potential appeared very reliable when compared to the experimental estimates of the inhibitory reversal potential and essentially excluded the occurrence of this excitatory conductance artifact.

The absolute values of inhibitory and excitatory conductance calculated by the analysis are contaminated with an error depending on the length constant of the neuron and the distance of the microelectrode from the synaptic inputs (*Calvin, 1969*; *Wehr and Zador, 2003*). Importantly, *Wehr and Zador, 2003* show that the estimate of the timing of synaptic conductances is reliable and not influenced by the length constant of the neuron. Therefore, the conductance profiles extracted with our analysis are qualitatively reliable, and the overall time course of inhibitory and excitatory conductances reflects the overall inhibitory and excitatory inputs to the different electrophysiological classes of active neurons analyzed.

## Implications of reconstructed synaptic conductance profiles for respiratory functional circuit architecture

Our analysis revealed different sets of inhibitory and excitatory synaptic conductances that are characteristic of the different populations of respiratory neurons. Indeed, our results suggest that phasic excitatory and inhibitory synaptic inputs that result from network interactions of these respiratory neuron populations shape membrane potential trajectories and spiking patterns of the neurons throughout the respiratory cycle. What can we learn from these results about the organization and operation of the underlying pattern generation networks?

We have suggested circuit motifs that are consistent with the patterns of synaptic inputs to the various neurons, and which also account for our immuno-histochemical identification of inhibitory neurons as subpopulations of most of the electrophysiological types. However, these circuit elements do not explain the overall dynamical operation of the circuits, particularly how the rhythm and the orderly generation of the multiple activity phases occurs. A more comprehensive synthesis of circuit interactions as illustrated below, and ultimately computational simulation studies (e.g. *Lindsey*

*et al., 2012*; *Molkov et al., 2017*), are required. In *Figure 4—figure supplement 1*, we illustrate the construction of a combinatorial circuit schematic of the functional connectivity from the neuronal population interactions suggested by our collection of circuit motifs.

Previously, computational models for core circuits of the respiratory CPG, including interacting microcircuits in the preBötC and BötC regions, have proposed a network organization with respiratory neuron population interactions that are capable from simulation studies with models incorporating circuit architectures (e.g. *Figure 4—figure supplement 1B*) and neuronal and synaptic biophysical properties to generate a realistic respiratory rhythm and the three-phase rhythmic pattern of neuron and neuronal population activity (*Lindsey et al., 2012*; *Molkov et al., 2017*; *Rubin et al., 2009*; *Rybak et al., 2007*). These include the model proposed by *Smith et al., 2007* based on regional circuit mapping and electrophysiological studies with the mature rat in situ experimental preparation used in the present studies. Despite its plausibility, this and other models remain speculative because the experimental evidence for functional connectivity is mostly indirect.

Our cellular-based analysis provides a way of untangling functional connections within the respiratory CPG and testing computational models. By assuming the origin of these inputs according to the phase in which they occur, we can draw conclusions on the functional connections between the key respiratory neuron populations recorded here. Our synaptic conductance profiles are consistent with multiple features of the network architecture and interactions proposed by *Smith et al., 2007* as illustrated in *Figure 4—figure supplement 1B*, which was suggested to be a minimal and sufficient circuit configuration to explain how the three-phase rhythmic respiratory pattern, including activity patterns of key interneuron populations in the medulla, can be generated. As reconstructed from our synaptic conductance profiles and collection of circuit motifs, there are new features in our more elaborate circuit connectome in the new combinatorial circuit schematic in *Figure 4—figure supplement 1A* as noted in the supplemental figure legend. There are likely additional connectivity features that remain to be uncovered as the catalogue of intracellular recording data and inferred functional interactions is expanded from further studies. We note that the profiles of neuronal synaptic interactions that we consider are specific for the in situ reduced rat brainstem-spinal cord system, which has proven to be an important model system for understanding the operation of core CPG respiratory circuits in the medulla and pontine reticular formation that are operational in our experimental set-up (*Paton et al., 2022*), but cannot account for the supra-brainstem and peripheral circuit inputs in the nervous system in vivo (e.g. *Lindsey et al., 2012*; *Smith et al., 2013*; *Paton et al., 2022*). Our objective here was to illustrate how our techniques can delineate the functional circuit interactions in an established core set of circuits in the respiratory CPG as an example of the utility of our approach that can be generally applied to rhythmic neural circuits as discussed below.

## Applicability to other rhythmically active networks

Our method for extracting and separating phasic patterns of inhibitory and excitatory synaptic conductances from single neuronal intracellular recordings is broadly applicable to rhythmic circuits, which are ubiquitous in nervous systems. Our approach involves phase-based analysis, where the network activity cycle is divided into activity phases to capture dynamic changes of synaptic input conductances throughout the activity cycle. This approach is relevant for any circuits with repetitive activity cycles, such as rhythmic CPG motor networks (e.g. *Calabrese and Marder, 2025*) and circuits generating cortical rhythms (*Yuste et al., 2005*). The method is based on the general mathematical framework presented, involving current balance equations and linear regression to separate excitatory and inhibitory conductances, and therefore is applicable to different rhythmic neuron types and circuits and is suitable for diverse experimental setups. By providing high-resolution temporal profiles of synaptic inputs and outputs (below, based on neuronal spiking patterns), this method enables researchers to infer the functional connectivity and dynamic interactions within any rhythmically active network.

Applying this method, as we have illustrated for respiratory circuits, to other rhythmic networks such as locomotor or cortical circuits, including sensory processing networks, follows a similar process for extracting neuronal excitatory and inhibitory inputs during different oscillatory phases of endogenous or sensory-driven periodic activity. This allows for the inference of functional synaptic input connections for any targeted neuron populations based on the timing and strength of the extracted synaptic inputs. In this context, functional populations are defined as groups of neurons with similar

firing patterns with respect to the cycle phase. If the synaptic conductance profile of a recorded neuron contains a component that mirrors the firing pattern of a specific population, it can be inferred that there is a functional connection from that population to the recorded neuron. Similarly, in sensory processing networks, it can reveal how sensory inputs are synaptically integrated in neurons by examining synaptic conductance patterns in response, for example, to repetitive stimuli.

The approach that we have described is versatile since it is compatible with various recording protocols, such as current-clamp and voltage-clamp, and different modes of intracellular recording (sharp electrode or whole-cell patch clamp). We note that our approach to defining functional connectivity does not provide information on the locations of the sources of the synaptic inputs, although we can infer spiking patterns of the source neurons from the temporal features of the synaptic conductance profiles. These profiles represent the total synaptic inputs that can arise from many neurons, possibly with short- and long-range axonal projections to a given neuron. The capability of our approach to read out the total synaptic conductance and separate the excitatory and inhibitory components from all sources at high temporal resolution is a strength of our approach. These inferred neuronal population connections from the electrophysiological analyses provided by our technique can be further mapped and corroborated with other functional and anatomical approaches.

In summary, the versatility of our technique lies in its foundation on basic electrophysiological principles and adaptability to different rhythmic activities, making it a powerful tool for investigating a wide range of rhythmically active neuronal networks and potentially offering insights into their functional organization and dynamic operations.

## Materials and methods

### Animal procedures

All animal procedures were approved by the Animal Care and Use Committee of the National Institute of Neurological Disorders and Stroke (Animal Study Proposal #1154–21).

### In situ perfused rat brainstem-spinal cord preparation

Experiments were performed using the in situ arterially perfused brainstem-spinal cord preparations from mature (3–4-wk-old) rats (*Paton, 1996*; *Paton et al., 2022*) following experimental procedures described in detail in *Smith et al., 2007* and modified as described below to provide optimal conditions for intracellular recording. Briefly, preheparinized (1000 units, given intraperitoneally) rats (Sprague-Dawley, 45–70 g; male) were anaesthetized deeply with 5% isoflurane, and the portion of the body caudal to the diaphragm was removed. The head and thorax were immersed in ice-chilled carbogenated Ringer solution, and the brain was decerebrated at a precollicular level. The descending aorta, phrenic nerve, central vagus, and hypoglossal nerves were surgically isolated. The dorsal brainstem was exposed by craniotomy and cerebellectomy. To achieve the mechanical stability necessary for intracellular recordings, the great veins and heart were removed. The preparation was transferred to a recording chamber, and the descending aorta was cannulated with a double-lumen catheter for perfusion and recording of perfusion pressure with a pressure transducer. A hydraulic damping Windkessel chamber was used in the perfusion circuit immediately downstream from the output of the roller pump (Watson-Marlow, Wilmington, MA) to eliminate peristaltic flow pulsations that affect the stability of the sharp electrode intracellular recordings.

### Solutions and pharmacological agents

The perfusate contained the following in distilled water: magnesium sulphate ($MgSO_4$–1.25 mM), potassium phosphate ($KH_2PO_4$–1.25 mM), potassium chloride (KCl–5.0 mM), sodium bicarbonate ($NaHCO_3$–25 mM), sodium chloride (NaCl–125 mM), calcium chloride ($CaCl_2$–2.5 mM), dextrose (10 mM), and polyethylene glycol (0.1785 mM). Vecuronium bromide was added to the perfusate to block neuromuscular transmission (4 µg/ml; SUN Pharmaceutical Industries, Bryan, USA). The perfusate was gassed with 95% $O_2$-5% $CO_2$ and maintained at 31 °C. Vasopressin (200–400 pM as required; APP Pharmaceuticals, East Schaumburg, USA) was added to the perfusate to raise and maintain perfusion pressure at 70–80 mmHg. Unless stated, all chemicals were from Sigma.

### Electrophysiological recordings and neuron labeling

Efferent activity of the phrenic (PN) and central vagus (VN) nerves was recorded simultaneously with glass suction electrodes and their activities were amplified, filtered (0.3–6.0 KHz), rectified, and

integrated (50 ms time constant). Sharp microelectrode somatic intracellular recordings were made in the preBötC and BötC regions. The brainstem was mapped by recording population activity of neurons in the medulla oblongata with an extracellular electrode (5–10 MOhm) prior to intracellular recording. The position of the electrode was monitored by software that kept track of the micromanipulator mechanical coordinates. Once the region of the preBötC or BötC was identified by rhythmic inspiratory/expiratory population activity, coordinates were saved, and in some cases, the brainstem surface was marked with pontamine sky blue dye. Intracellular recordings were made with fine-tipped glass microelectrodes (45–70 MOhm) pulled from borosilicate glass capillaries. The microelectrodes for intracellular recording were driven into the recording site by a combination DC motor-driven micromanipulator with an integrated piezotranslator (PM10-1, Marzhauser Wetzlar, Germany), allowing high stepping speeds for cell penetration. A dual current clamp-voltage clamp amplifier (SEC-05X, NPI, Tamm, Germany) was used to amplify the signals in bridge and discontinuous current clamp (DCC) recording modes. Discontinuous single electrode voltage clamp (SEVC) recordings were obtained by exploiting the high switching frequencies and electrode capacity compensation of the SEC-05X system, allowing high-fidelity SEVC recordings with the high-resistance sharp microelectrodes used. The microelectrodes were filled with a solution of 2 M potassium acetate (KAc) and 0.1 M potassium chloride (KCl), and 4% neurobiotin. During recordings, some neurons were labeled by ionophoresis of neurobiotin into the cell. All recordings were digitized via a CED 1401-plus multi-channel interface and acquired (6–12 kHz digitization rates) using CED Spike 2 software.

## Histological analysis

Transected brain stem was fixed in 4% buffered (0.1 M phosphate buffer) paraformaldehyde for 24 hr at 4 °C, cryoprotected at 4 °C in 30% sucrose, and subsequently coronally sectioned (30 μm thick) on a freezing microtome. All the floating sections were preincubated for 1 h with 10% donkey serum in PBS and subsequently incubated for 48–72 hr at room temperature with primary antibodies: Rabbit anti-Glycine antibody (1:5000; Immunosolution), rabbit anti-GABA antibody (1:1000; Millipore-Sigma), and goat anti-ChAT antibody (1:500; Chemicon). Labeling for neuronal ChAT expression was used to label motoneurons of the nucleus ambiguous to verify the region where the neurons were recorded and to confirm that the recordings were obtained from non-motoneurons. Floating sections were incubated for 2 hr at room temperature with a mixture of donkey anti-rabbit IgG conjugated with Cy5 and donkey anti-goat IgG conjugated with Cy3 (1:500; Jackson ImmunoResearch). For fluorescence labeling of neurons filled with neurobiotin during electrophysiological recording, slices were incubated with Alexa Fluor 488-Avidin D (1:500, Invitrogen). Tissue was mounted on slides and covered with antifading medium. Confocal images were obtained with a laser-scanning microscope (LSM 510 Meta, Zeiss). Alexa Fluor 488, Cy3, and Texas Red or Cy5 were detected at 488, 543, and 633 nm emission wavelengths, respectively. Images were acquired and processed with the Zeiss Zen software and presented as single optical sections selected from serial optical sections (Z-stacks).

## Reconstruction of synaptic inputs from intracellular recordings
### Preprocessing
Inhibitory and excitatory synaptic inputs to respiratory neurons were reconstructed from intracellular recordings using a general approach similar to that described by *Berg et al., 2007* and others (*Borg-Graham et al., 1998*; *Anderson et al., 2000*; *Shu et al., 2003*; *Endo and Kiehn, 2008*).

During intracellular recordings for conductance measurement and subsequent decomposition of inhibitory and excitatory inputs, depolarizing and hyperpolarizing current was injected into the cell in a stepwise manner (*Figure 1A*), and data at hyperpolarized voltages (–100 to –60 mV) was used for the analyses. The period of time from the onset of the phrenic burst to the beginning of the next phrenic burst was used as the reference cycle. The beginning of a phrenic burst was determined as 10% of phrenic amplitude in the rectified and low- pass filtered (time constant = 0.05 s) extracellular recording. We only analyzed recordings from respiratory neurons that fulfilled the following criteria: (1) The coefficient of variation of the respiratory cycle period was <10%; (2) At least three steps of current injections were made during intracellular recording that were kept constant for at least 5 respiratory cycles each. Neurons with obvious recording instability, such as significant changes in membrane leakage current with a baseline membrane potential drift were not analyzed. The intracellularly recorded membrane potential $V_m(t)$ was sampled down from 12.5 kHz to 100 Hz. To eliminate

spikes without affecting slow membrane potential fluctuations, the voltage trace $V_m(t)$ was processed using median filtration.

## Error estimation

After performing linear regression (2) for each phase bin, we have standard errors $\delta R(\varphi)$ and $\delta V_0(\varphi)$ of the total resistance $R(\varphi)$ and the resting potential $V_0(\varphi)$, respectively, which are defined by voltage fluctuations and by the number of respiratory cycles in the epoch used for reconstruction. Based on this, we can approximately calculate the standard error for the total conductance $\delta G(\varphi) = \delta R(\varphi)/R^2(\varphi)$, which can be used as an overestimate for the errors of both excitatory and inhibitory conductances since the total conductance is a sum of the two. Calculation of the dynamic components involves subtraction of the minimal values of the calculated synaptic conductances. Therefore, the dynamic components' standard errors can be estimated as $\delta\Delta G_{i,e}(\varphi) = \sqrt{\delta G(\varphi)^2 + \delta G(\varphi_m)^2}$, where $\varphi_m$ is the phase where the corresponding conductance (excitatory or inhibitory) attains its minimum.

## Statistical significance

Considering the relatively large sample sizes (for each cell analyzed we had 50–200 respiratory cycles long recordings), we used a one-tailed z-test to confirm statistical significance of the reconstructed synaptic inputs by testing a null hypothesis that the dynamical component of the synaptic conductance is zero (no different from its minimal value). Specifically, at each phase of the respiratory cycle $\varphi$ for the dynamic components of inhibitory and excitatory conductances, we calculated their z-scores $Z_{i,e}(\varphi) = \Delta G_{i,e}(\varphi)/\delta\Delta G_{i,e}(\varphi)$, and corresponding p-values. We used p-value <0.05 criterion to accept the alternative hypothesis that the calculated dynamic component of the synaptic conductance at a particular phase is statistically significantly greater than zero. In the latter case, we concluded that the recorded cell received the corresponding type of synaptic input (excitatory or inhibitory) during the corresponding part of the respiratory cycle (inspiration, post-inspiration, or late-expiration/pre-inspiration).

# Acknowledgements

This study was supported by NIH grants R01 NS057815, R01 NS069220, R01 AT008632, the Alexander von Humboldt Foundation, and the Intramural Research Program of the NIH, NINDS.

# Additional information

### Competing interests

Jeffrey Smith: Reviewing editor, eLife. The other authors declare that no competing interests exist.

### Funding

| Funder | Grant reference number | Author |
| --- | --- | --- |
| National Institutes of Health | R01 NS057815 | Yaroslav Molkov Jeffrey C Smith |
| Alexander von Humboldt Foundation | | Anke Borgmann |
| National Institutes of Health | R01 AT008632 | Yaroslav Molkov |
| Intramural Research Program of the NIH, NINDS | | Jeffrey Smith |

| Funder | Grant reference number | Author |
|---|---|---|

The funders had no role in study design, data collection and interpretation, or the decision to submit the work for publication.

## Author contributions

Yaroslav Molkov, Conceptualization, Resources, Data curation, Software, Formal analysis, Supervision, Funding acquisition, Validation, Investigation, Visualization, Methodology, Writing – original draft, Project administration, Writing – review and editing; Anke Borgmann, Hidehiko Koizumi, Data curation, Formal analysis, Validation, Investigation, Methodology, Writing – review and editing; Noriyuki Hama, Data curation, Formal analysis, Investigation, Methodology, Writing – review and editing; Ruli Zhang, Data curation, Visualization, Methodology; Jeffrey Smith, Conceptualization, Resources, Data curation, Formal analysis, Supervision, Funding acquisition, Validation, Investigation, Methodology, Writing – original draft, Project administration, Writing – review and editing

## Author ORCIDs

Yaroslav Molkov https://orcid.org/0000-0002-0862-1974
Hidehiko Koizumi https://orcid.org/0000-0002-7747-3434
Jeffrey Smith https://orcid.org/0000-0002-7676-4643

## Ethics

All animal procedures were approved by the Animal Care and Use Committee of the National Institute of Neurological Disorders and Stroke (Animal Study Proposal #1154-21).

Reviewer #2 (Public review): https://doi.org/10.7554/eLife.101959.4.sa1
Author response https://doi.org/10.7554/eLife.101959.4.sa2

---

# Additional files

## Supplementary files

MDAR checklist

## Data availability

All data generated as well as software developed to analyze it during this study are available on Dryad, DOI: https://doi.org/10.5061/dryad.bcc2fqzrp.

The following dataset was generated:

| Author(s) | Year | Dataset title | Dataset URL | Database and Identifier |
|---|---|---|---|---|
| Molkov YI, Borgmann A, Koizumi H, Hama N, Zhang R, Smith J | 2025 | Inference technique for the synaptic conductances in rhythmically active networks and application to respiratory central pattern generation circuits | https://doi.org/10.5061/dryad.bcc2fqzrp | Dryad Digital Repository, 10.5061/dryad.bcc2fqzrp |

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
