## [Editor Report · eLife Assessment]

This work describes an inference technique for extracting information about relative contributions of excitatory and inhibitory synaptic drive onto single neurons in neural networks. The electrophysiological techniques and results are of high quality, and the analytical work is novel and potentially powerful, yet with several untested assumptions underlying the approach. This is nevertheless **solid** work that will be **valuable** to neuroscience labs interested in exploring alternative approaches to studies of integrated synaptic connectivity.

---

## [Referee Report · Reviewer #2 (Public review)]

Summary:

By measuring intracellular changes in membrane voltage from a single neuron of the medulla the authors attempted to develop a method for determining the balance of excitatory and inhibitory synaptic drive onto a single neuron.

Strengths:

This data-driven approach to explore neural circuits is described well in this study and could be valuable in identifying microcircuits that generate rhythms. Importantly, perhaps, this inference method could enable microcircuits to be studied without the need for time-consuming anatomical tracing or other more involved electrophysiological techniques. Therefore, I can see the value in developing an approach of this type.

Weaknesses:

The implications of several assumptions associated with this inference technique have been considered by the authors.

Most importantly, it is my understanding that this approach assumes a linear I-V when extracting information about the excitatory and inhibitory synaptic conductances (see equations 6 and 7). In Figure 6, the authors explore the impact of varying the reversal potential for the extraction of information about synaptic drive, but this still assumes that the underlying conductance is linear. However, open rectification will be a feature of any conductance generated by asymmetric distributions of ions (see the GHK current equation) and will therefore be a particular issue for the inhibition resulting from asymmetrical Cl- ion gradients across GABA-A receptors as well as the K+ conductance indirectly activated by GABA-B receptor activation. The mixed cation conductance that underlies most synaptic excitation will also generate a non-linear I-V relationship due to the inward rectification associated with polyamine block of AMPA receptors. The authors present evidence that the I-V relationship is linear over most of the voltage range examined, and this is a helpful addition. The authors have discussed the absence of active conductances contributing to the I-V, but I still wonder how the extraction of information concerning the excitatory and inhibitory conductances relies on the assumption of a linear I-V for these conductances.

This approach has similarities to earlier studies undertaken in the visual cortex that estimated the excitatory and inhibitory synaptic conductance changes that contributed to membrane voltage changes during receptive field stimulation. However, these approaches also involved the recording of transmembrane current changes during visual stimulation that were undertaken in voltage-clamp at various command voltages to estimate the underlying conductance changes. Molkov et al have attempted to essentially deconvolve the underlying conductance changes without this information and I am concerned that this simply may not be possible. However, I appreciate the efforts taken by the authors to address this issue.

The current balance equation (1) cited in this study is based upon the parallel conductance model developed by Hodgkin & Huxley. One key element of the HH equations is the inclusion of an estimate of the capacitive current generated due to the change in voltage across the membrane capacitance. While the present study considers the impact of membrane capacitance, a deeper discussion on how variations in capacitance across different neuron types might affect inference accuracy would be useful. Differences in capacitance could introduce variability in inferred conductances, potentially influencing model predictions.

Studies using acute slicing preparations to examine circuit effects have often been limited to the study of small microcircuits, especially feedforward and feedback interneuron circuits. It is widely accepted that any information gained from this approach will always be compromised by the absence of patterned afferent input from outside the brain region being studied. In this study, descending control from the Pons and the neocortex will not be contributing much to the synaptic drive and ascending information from respiratory muscles will also be absent completely. This may not have been such a major concern if this study had been limited to demonstrating the feasibility of a methodological approach. However, this limitation does need to be considered when using an approach of this type to speculate on the prevalence of specific circuit motifs within the medulla (Figure 4). Therefore, I would argue that some discussion of this limitation should be included in this manuscript.

---

## [Author Response]

The following is the authors’ response to the previous reviews.

**Reviewer #1 (Public Review):**
Comments on revisionsThe authors have done a good job at revising the manuscript to put this work into the context of earlier work on brainstem central pattern generators.

Thank you.

I still believe the case for the method is not as convincing as it would have been if the method had been validated first on oscillations produced by a known CPG model. Why would the inference of synaptic types from the model CPG voltage oscillations be predetermined? Such inverse problems are quite complicated and their solution is often not unique or sufficiently constrained. Recovering synaptic weights (or CPG parameters) from limited observations of a highly nonlinear system is not warranted (Gutenkunst et al., Universally sloppy parameter sensitivities in systems biology models, PLOS Comp. Biol. 2007; http://www.doi.org/10.1371/journal.pcbi.0030189) especially when using surrogate biological models like Hodgkin-Huxley models.

The model of the CPG is irrelevant for such a test of validity because what we reconstruct are postsynaptic conductances of an individual neuron. The network creates a periodic input to this neuron and thus forms a periodic pattern of excitatory and inhibitory conductances. The nature of this input, whether autonomously generated or created artificially (say by periodic optogenetic stimulation), is generally not important. To illustrate this, we used a one-compartment conductance-based (Hodgkin-Huxley style) model neuron incorporating a certain common set of channels (fast sodium (*INaF*), potassium delayed rectifier (*IKdr*), persistent sodium (*INaP*), calcium-dependent potassium (*IKCa*), and cationic non-specific current (*ICAN*)), as well as excitatory and inhibitory synaptic channels whose conductances were implemented as predefined periodic functions. The test suggested by the reviewer would be to implement a current-step protocol similar to the experiments and apply our technique to see if the reconstructed conductance profiles match those predefined functions. Below we show the reconstruction steps for the following arbitrarily chosen pattern:

𝑔_𝐸𝑋𝐶_(𝑡) /𝑔_𝐿𝐸𝐴𝐾_ = 0.1(1 + sin(π𝑡)) and 𝑔_𝐼𝑁𝐻_(𝑡)/𝑔_𝐿𝐸𝐴𝐾_ = 0.1 (1 + cos(π𝑡)). Author response image 1 below shows the baseline activity of this model neuron in the absence of the injected current.

**Author response image 1. sa2fig1:** 

Then we applied a current-step protocol with four steps producing different levels of hyperpolarization and applied our method by calculating the total conductance using linear regression (see the current-voltage plots below) and then decomposing it into the excitatory and inhibitory components.

As one can see, the reconstructed conductances in Author response image 3 below are nearly identical to their theoretical profiles. This is not surprising because all voltage-dependent currents in the model neuron were inactive in the range of voltages matching our experimental conditions. Therefore, the model could be reduced to just the leak current, synaptic currents and the injected current, which matches precisely the model we used in our manuscript.

**Author response image 3. sa2fig3:** 

In p.2, the edited section refers to the interspike interval being much smaller than the period of the network. More important is to mention the relationship between the decay time of inhibitory synapses and the period of the network.

This interpretation misunderstands the focus of our method. The edited sections (including in the theory section of Results) highlight the conditions under which the capacitive current becomes negligible, emphasizing that the membrane time constant must be much smaller than the network oscillation period. This separation of time scales ensures that the membrane potential adjusts quickly to changes in postsynaptic conductance, rendering the capacitive current insignificant over the network’s rhythm. In contrast, the synaptic decay time governs how presynaptic inputs are transduced into postsynaptic conductances—a process relevant to understanding synaptic dynamics but not directly tied to our method’s core objective. Our approach reconstructs postsynaptic conductances from intracellular recordings, not presynaptic spike trains. While interpreting these conductance profiles in terms of specific synaptic connections would indeed involve synaptic decay dynamics, such an analysis exceeds the scope of our paper. Thus, the condition emphasized in the edited sections—concerning the membrane time constant and network period—is the critical one for our method’s applicability, and the synaptic decay time, while relevant to broader synaptic modeling, does not undermine our conclusions.

We have added the requirement for a much smaller membrane time constant in the Introduction on page 2. The Results theory section already incorporates an extensive discussion of this requirement.

**Comments from the editors:**
We apologize for the delay in coming to this decision, but there was quite a bit of post-review discussion that needed to be resolved. There are two issues that the reviewers agree should be addressed. They remain unconvinced that the simplifying assumptions of the approach are valid. (1) The main issue with the phase argument is that the biological synaptic conductance depends on time and not on the phase of the respiratory cycle as mentioned in the first round of reviews. The approximation g(t)=g(phase) seems to be far too simple to be biologically realistic.

As we elaborate below, time and phase are fundamentally and mathematically equivalent representations of the same underlying dynamics in a periodic system, and thus, a phase-based representation—where conductances are expressed as functions of the cycle’s phase—is a justified and effective approach for capturing their behavior. We have added this explanation to the theory section of Results. Below are the bases for our assertion.

In a periodic system, such as the respiratory CPG, the system’s behavior repeats at regular intervals, defined by a period T. For the respiratory cycle in our experimental preparation, this period is approximately 3–4 seconds, encompassing phases like inspiration, post-inspiration, and expiration. In such systems:

Time (t) is a continuous variable that progresses linearly.

Phase (φ) represents the position within one cycle, typically normalized between 0 and 1 (or 0 to 2π in some contexts). It can be mathematically related to time via: φ(t) = (t mod T)/T, where (t mod T) is the time elapsed within the current cycle.

Because the system is periodic, any variable that repeats with period T—such as synaptic conductance in a rhythmically active network—can be expressed as a function of either time or phase. Specifically, if g(t) is periodic with period T, then g(t) = g(t+T). This periodicity allows us to redefine g(t) in terms of phase: g(t) = g(φ(t)), where φ(t) maps time onto a repeating cycle. Thus, in a periodic system, time and phase are fundamentally equivalent representations of the same underlying dynamics. Saying that synaptic conductance depends on phase is mathematically equivalent to saying it depends on time in a periodic manner.

In a rhythmically active network like the respiratory central pattern generator (CPG), the synaptic conductances, regardless of the specific mechanisms by which they are formed, exhibit periodicity that matches the network’s oscillatory cycle. This occurs because the conductances are driven by the repetitive activity of presynaptic neurons, which are synchronized to the network’s overall rhythm. As a result, the synaptic conductances vary with the same period as the network, making a phase-based representation—where conductances are expressed as functions of the cycle’s phase—a justified and effective approach for capturing their behavior. In our study, we utilized the in situ arterially perfused brainstem-spinal cord preparation from mature rats, which is known to produce a highly periodic respiratory rhythm. To ensure the consistency of this periodicity, we carefully selected recordings where the coefficient of variation of the respiratory cycle period was less than 10%, as outlined in our methods. This strict selection criterion confirms the stability and regularity of the rhythm, supporting the validity of using a phase representation to analyze the synaptic conductances.

(2) Figure S1 is problematic. First, the currents injected appear to be infinitesimally small.

There was a typo in the current units, which should be nA and not pA, as evident from the injected current–membrane potential plots in Figure 1B. Figure S1 has been corrected.

Second, the input resistance is completely independent of voltage, as though there was little or no contribution from hyperpolarization activated currents, which would be surprising.

While hyperpolarization-activated currents are indeed present in many neuronal types and could theoretically affect input resistance, our data consistently show linear I-V relationships across the voltage range tested (-60 to -100 mV) for the neurons analyzed (see Figure S1 and Author response image 4-9 below). This linearity suggests that, under our experimental conditions, the contribution of voltage-dependent currents, such as h-currents, is negligible within this range.

Additionally, we now indicate in the manuscript in the theory section of Results how the presence of significant hyperpolarization-activated h-currents would impact our synaptic conductance reconstruction method. In current-clamp recordings, non-linearity from h-currents could introduce voltage-dependent changes in total conductance unrelated to synaptic inputs, potentially skewing the reconstruction. However, this concern does not apply to voltage-clamp recordings, where the membrane potential is held constant, eliminating contributions from voltage-dependent intrinsic currents. As strong evidence of the minimal influence of h-currents, we directly compared synaptic conductance reconstructions using both current-clamp and voltage-clamp protocols in a subset of neurons. The results from these two approaches were highly consistent, indicating that h-currents do not significantly affect our findings. This robustness across experimental methods reinforces the reliability of our conclusions.

Together, the linear I-V relationships and the agreement between current- and voltage-clamp reconstructions provide compelling evidence that our method accurately captures synaptic conductances without interference from h-currents.

Typical examples of I-V relationships for each respiratory neuron firing phenotype:

**Author response image 4. sa2fig4:** ramp-I.

**Author response image 5. sa2fig5:** pre-I/I.

**Author response image 6. sa2fig6:** post-I.

**Author response image 7. sa2fig7:** aug-E.

**Author response image 8. sa2fig8:** early-I.

**Author response image 9. sa2fig9:** late-I.